# Long-acting injectable atovaquone nanomedicines for malaria prophylaxis

Rahul P. Bakshi[1,2], Lee M. Tatham[3], Alison C. Savage[4], Abhai K. Tripathi[2,5], Godfree Mlambo[2,5], Matthew M. Ippolito[1,2,6], Elizabeth Nenortas[1,2], Steve P. Rannard[4], Andrew Owen [3] & Theresa A. Shapiro[1,2,5]

Chemoprophylaxis is currently the best available prevention from malaria, but its efficacy is compromised by non-adherence to medication. Here we develop a long-acting injectable formulation of atovaquone solid drug nanoparticles that confers long-lived prophylaxis against *Plasmodium berghei* ANKA malaria in C57BL/6 mice. Protection is obtained at plasma concentrations above 200 ng ml$^{-1}$ and is causal, attributable to drug activity against liver stage parasites. Parasites that appear after subtherapeutic doses remain atovaquone-sensitive. Pharmacokinetic–pharmacodynamic analysis indicates protection can translate to humans at clinically achievable and safe drug concentrations, potentially offering protection for at least 1 month after a single administration. These findings support the use of long-acting injectable formulations as a new approach for malaria prophylaxis in travellers and for malaria control in the field.

[1] Division of Clinical Pharmacology, Departments of Medicine and of Pharmacology and Molecular Sciences, The Johns Hopkins University, 725 North Wolfe Street, Baltimore, MD 21205, USA. [2] The Johns Hopkins Malaria Research Institute, Baltimore, MD 21205, USA. [3] Department of Molecular and Clinical Pharmacology, University of Liverpool, Block H, 70 Pembroke Place, Liverpool L69 3GF, UK. [4] Department of Chemistry, University of Liverpool, Crown Street, Liverpool L69 7ZD, UK. [5] Department of Molecular Microbiology and Immunology, The Johns Hopkins University, Baltimore, MD 21205, USA. [6] Division of Infectious Diseases, Department of Medicine, The Johns Hopkins University, Baltimore, MD 21205, USA. Steve P. Rannard, Andrew Owen and Theresa A. Shapiro contributed equally to this work. Correspondence and requests for materials should be addressed to S.P.R. (email: srannard@liverpool.ac.uk) or to A.O. (email: aowen@liverpool.ac.uk)

Every year, *Plasmodium falciparum* malaria afflicts hundreds of millions of people and kills hundreds of thousands of children[1]. Despite considerable success in reducing the worldwide prevalence of malaria, its incidence in visitors to endemic areas has continued to rise steadily[2, 3], and for unprotected travellers to west Africa the risk of malaria is 3.4% mo$^{-1}$ [4]. Though promising advances are being made, as of yet no malaria vaccine can reliably impart high levels of long-lived protection. Antimalarial drugs thus continue to provide an essential component of malaria prophylaxis. Unfortunately, however, the requirement for daily or weekly oral drug-dosing regimens is commonly associated with noncompliance[2, 3].

Human infection with falciparum malaria begins with sporozoite inoculation by a female anopheline mosquito. This parasite form migrates rapidly to the liver, invades hepatocytes and amplifies ~20,000-fold, then spills into the bloodstream to initiate the self-propagating 48 h erythrocytic cycle responsible for all disease manifestations (Fig. 1). Most antimalarial drugs target erythrocytic parasites, but several are effective against liver stages. To preclude resistance, ill patients (who may harbour up to $10^{12}$ infected red cells) are treated with drug combinations. However, in the numerically advantageous situation of malaria prophylaxis, where drug concentrations are well established before the introduction of a small number of parasites, monoprophylaxis (e.g., with mefloquine or doxycycline) is a common and highly effective option[4].

Atovaquone (Fig. 2a), a structural mimic of ubiquinone, selectively binds to the cytochrome b of malaria parasites, inhibits respiration and collapses the trans-mitochondrial membrane potential[5, 6]. A broad-spectrum antiprotozoal, for 25 years orally dosed atovaquone has been used for the prophylaxis and treatment of *Pneumocystis* pneumonia in immunosuppressed patients. In this setting, its approved dosing to obtain sustained and high target plasma concentrations (above 20 µg mL$^{-1}$, 54 µM)[7] is safe and well tolerated[5]. Atovaquone is not metabolised, exhibits few drug–drug interactions, and has a plasma half-life measurable in days. It has potent sub-nanomolar activity against erythrocytic *P. falciparum* in vitro, is not cross-resistant with other antimalarials, and has activity against both liver stage (causal) and erythrocytic (suppressive) parasites. A causal mechanism of prophylaxis has several distinct advantages, including the fact that it targets a smaller number of parasites (before they complete a four log amplification in liver) and clinically that it requires just 1,

not 4, weeks of postexposure dosing. To preclude resistance in patients with established erythrocytic infection, atovaquone was marketed in fixed combination with proguanil. However, recent studies have indicated that atovaquone-resistant *P. falciparum* cannot survive in, or be transmitted by, mosquitoes[8]. This remarkable phenomenon would not affect the risk of resistance in atovaquone monotherapy of established erythrocytic parasitemia, but it does avoid the problem of monoprophylaxis failure stemming from mosquito-borne transmission of atovaquone-resistant parasites.

Most approaches for making nanoparticle formulations involve creating either drug-associated nanocarriers (drug conjugation, complex-formation or encapsulation)[9, 10] or excipient-stabilised nanoparticles of drug (solid drug nanoparticles; SDNs)[11–13], using attrition methods such as nanomilling or high-pressure homogenisation. A major limitation of both approaches is the low ratio of drug to non-therapeutic excipients that can be attained (commonly ≤25% dry weight). Emulsion-templated freeze drying (ETFD)[14, 15] is an alternative method for creating SDNs that entails the use of inactive commercial excipients commonly used in FDA-approved medicines, and allows very high-drug loadings relative to excipients (often ≥70% by weight). Orally administered high-drug content SDN formulations, initially identified through ETFD screening and latterly scaled using emulsion spray drying under good manufacturing practices conditions, are now in clinical trials for antiretroviral indications[16, 17]. The approach has also been employed to determine how physical characteristics impact biological performance by screening nanoformulation libraries[18, 19]. Nano-milled antiretroviral drugs have recently received considerable attention after clinical trials of LAI (long-acting injectable) medicines demonstrated therapeutic plasma drug exposure for between 1 and 3 months, offering treatment simplification and addressing medication adherence[20–22]. Importantly, these formulations have also shown remarkable preclinical utility for pre-exposure prophylaxis for which they are now in clinical trials, validating the LAI approach for prophylaxis. The marked hydrophobicity of atovaquone (log $P = 4.7$) and its low systemic clearance make it particularly suitable for LAI. Moreover, the high-drug content of ETFD-generated formulations minimises the required injection volume and its associated intolerability.

Here, using the stringent *P. berghei* ANKA-C57BL/6 mouse model for malaria, we have found that intramuscular atovaquone

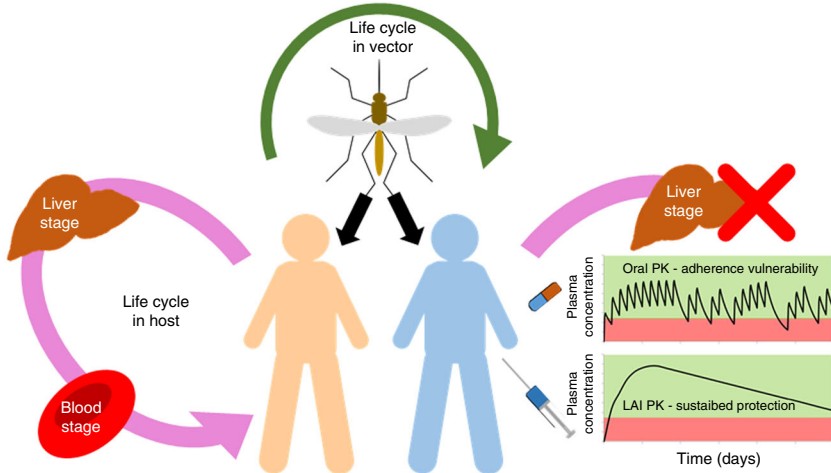

**Fig. 1** Utility of long-acting antimalarial prophylaxis. Pharmacokinetic exposure to atovaquone monotherapy via oral administration blocks liver and erythrocytic stages of the parasite life cycle within the host (causal and suppressive activity, respectively). However, there is a vulnerability of oral dosing to non-adherence. The current work reports the preclinical development of an intramuscular long-acting nanomedicine, which provides sustained protection to parasite exposure in a preclinical model, expected to provide at least 1-month protection in humans

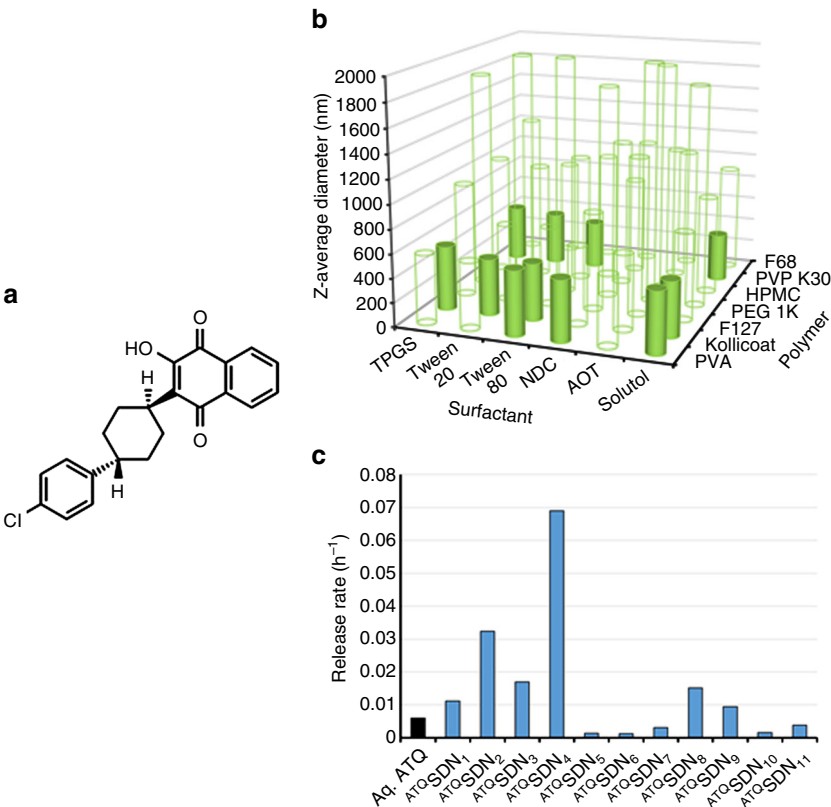

**Fig. 2** Creation and screening of atovaquone nanoformulations. **a** Hydrophobic hydroxynaphthoquinone atovaquone was formulated to generate candidate ATQSDN nanoparticles. **b** ATQSDN library was screened for Z-average nanoparticle diameter after dispersion into water (0.5 mg mL⁻¹, 25 °C); 'hits' (solid bars) were selected for further evaluation. **c** Eleven ATQSDN 'hits' were evaluated using an in vitro rapid equilibrium dialysis intramuscular depot model assay (SIF) using ATQSDN formulation prepared with ³H-labelled atovaquone

nanoformulation provides reliable chemoprophylaxis, for up to 4 weeks after dosing, against an intravenous sporozoite challenge. Pharmacokinetic–pharmacodynamic analyses indicate that protection is obtained at plasma atovaquone levels well below those currently in clinical use for other indications, and that protection is attributable to causal action against liver stage parasites.

## Results

**Preparation and characterisation of nanoparticle library.** A library of SDNs, containing atovaquone and a binary combination of polymer and surfactant excipient stabilisers, was created using the recently reported rapid ETFD-screening approach[16]. Seven candidate polymers and six surfactants were chosen from the FDA CDER list of inactive ingredients[23] and the resulting 42 formulations, all 80 wt% atovaquone, 13 wt% polymer and 7 wt% surfactant, were assessed by dynamic light scattering (Fig. 2b). Candidate formulations were chosen for progression if the following criteria were met: (1) complete aqueous dispersion at 0.5 mg mL⁻¹; (2) SDN Z-average diameter <1000 nm to match existing LAI paradigms; (3) dynamic light-scattering analysis <5% SD (three scans) within the measured Z-average diameter indicating stability during analysis; and (4) a measured polydispersity index <0.4 to maximise uniformity of drug release from injected SDNs. All 42 candidates were ranked against these criteria and eleven (Supplementary Table 1) were identified that also passed the reproducibility screening in subsequent repeated syntheses (Supplementary Fig. 1). SDN zeta potentials for the eleven candidates ranged from +8 to −20 mV (Supplementary Table 1), indicating that stability of the dispersed samples in water was predominantly correlated to steric stabilisation derived

from the water-soluble excipients. ³H-atovaquone utilised in a miniaturised ETFD SDN production and characterisation revealed very strong correlation of outcomes at this reduced scale (Supplementary Fig. 1).

**Atovaquone release from SDN formulation candidates.** The 11 SDN candidates were studied by in vitro rapid equilibration dialysis. Each SDN was dispersed into simulated interstitial fluid (SIF) and compared with unformulated atovaquone (<1% dimethyl sulfoxide (DMSO) in SIF). Scintillation counting allowed rapid determination of drug release, and five candidates (ATQSDN₂, ₄, ₆, ₇, ₈) providing a 50-fold range of drug-release rates (Fig. 2c) were progressed to in vivo testing. High-drug content minimises depot volume but may also negatively impact the desired release rate. Therefore, SDN formulations were generated for ATQSDN₄, ATQSDN₆ and ATQSDN₈ (that exhibited highest, lowest and intermediate release rate in initial screens, respectively) using the same procedures and excipient mixtures, to establish the effect on release rate of systematic lowering of the drug loading from 80 wt% through to 20 wt% (Supplementary Table 2, and Supplementary Figs. 2–5). For all three formulations, a decrease in release rate was observed as drug content increased to 60 wt%. Although this trend was continued to 80 wt% for ATQSDN₆ and ATQSDN₈ ATQ. A higher release rate for ATQSDN₄ was observed at 80 wt% drug loading. These data highlight a critical need to determine release kinetics at different drug loadings for LAI.

***P. berghei* ANKA-C57BL/6 in mice.** This model is widely used for preclinical evaluation of therapies directed against

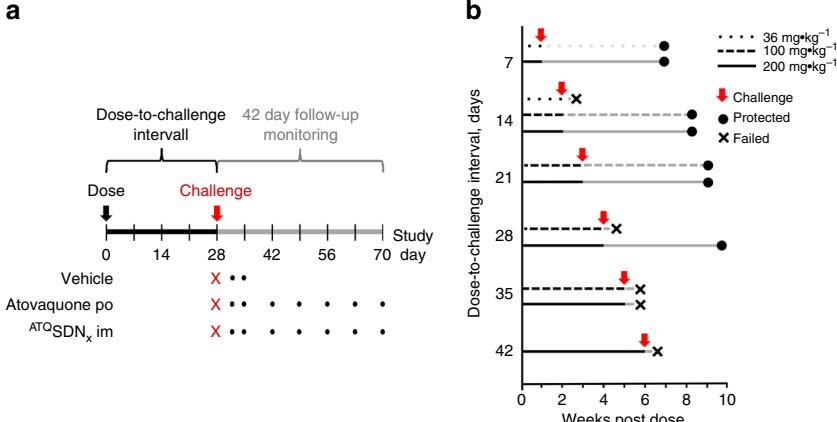

**Fig. 3** Efficacy testing of $^{ATQ}SDN_7$. **a** Experimental scheme. Mice dosed on day 0 with placebo, oral atovaquone or intramuscular nanoparticle atovaquone formulation were challenged once with intravenous *P. berghei* sporozoites at a given interval after dosing (28 days is depicted). For 42 days after challenge, blood samples were obtained and monitored for parasitemia (dots). **b** Prophylactic efficacy of intramuscular $^{ATQ}SDN_7$. Cohorts of mice treated on day 0 with indicated doses of $^{ATQ}SDN_7$ were challenged at a given interval after dosing (red arrows). Black lines, dose-to-challenge interval; grey lines, 42 days follow-up monitoring period. Prophylaxis was successful (circles) if, in two independent experiments (each with a cohort of 3–5 mice) all animals remained parasite-free for 42 days after challenge. Prophylaxis failed (x) if patent parasitemia was detected in any mouse in the cohort. For failed regimens the actual intervals between challenge and failure are not depicted. Not shown, all concurrent placebo recipients developed parasitemia, and all concurrent oral atovaquone controls failed challenge on or before 7 days after dosing. In all comparisons, successful prophylaxis with intramuscular $^{ATQ}SDN_7$ was superior to no-drug control at $p \leq 0.003$ (Fisher's exact test)

*P. falciparum*[24]. Infection in this system is lethal, and parasitemia reliably results from inoculation of just 50 sporozoites. However, there are notable differences between *falciparum* and *berghei*, including the shorter development times for *P. berghei* in liver (48 h vs 6.5 d) and erythrocytes (24 vs 48 h). In our hands over 14 months' time, in experiments involving some 320 mice, an intravenous challenge of 5000 sporozoites was lethal in every untreated animal, and the liver stage was confirmed to require $\geq 45$ and $\leq 48$. In all animals that developed patent malaria (regardless of treatment), parasites appeared $5.5 \pm 3.0$ days (M ± SD, 177 mice) after sporozoite challenge. The longest prepatent period was 21 days (in an animal last tested on day 11), and the interval between patency and death was $6.9 \pm 3.5$ days (177 mice).

To assess prophylactic activity, mice were injected once with atovaquone, challenged once with sporozoites, and then monitored 42 days for patent erythrocytic parasitemia (Fig. 3a). The conservative 6-week follow-up interval more than eclipsed the 21 days longest prepatent period that we saw. Prophylaxis was deemed successful only if no animal in a dosing cohort became parasitemic by 42 days after challenge, in two independent experiments. Experiments included untreated controls to validate the challenge, and mice treated with the same dose of atovaquone given orally.

In a preliminary survey, all five candidate nanoformulations afforded some protection when injected 1 day before challenge, but only $^{ATQ}SDN_7$ and $^{ATQ}SDN_8$ were fully protective against sporozoite infusion at 7 days after dosing, so these formulations were taken forward for further testing. Mice injected with 36, 100 or 200 mg kg$^{-1}$ $^{ATQ}SDN_7$ were challenged at intervals up to 6 weeks after dosing (Fig. 3b). For $^{ATQ}SDN_7$ or $^{ATQ}SDN_8$, the efficacy (and statistical analyses) were the same: protection for up to four weeks after challenge. In these experiments, oral atovaquone at doses up to 200 mg kg$^{-1}$ failed to protect against challenge at $\leq 7$ days after dosing, as did intramuscular polymer plus surfactant (nanoformulation 'vehicles') at concentrations equivalent to those in a 200 mg kg$^{-1}$ dose of $^{ATQ}SDN_7$ or $^{ATQ}SDN_8$.

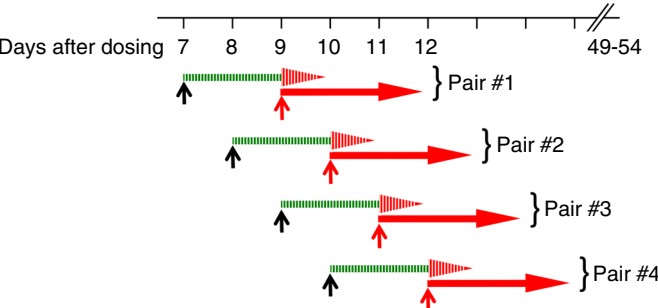

**Fig. 4** Causal vs suppressive prophylactic antimalarial activity. On day 0 mice were injected with 36 mg kg$^{-1}$ $^{ATQ}SDN_7$, then four paired cohorts (three mice per cohort) were challenged at the indicated times with 5000 *P. berghei* sporozoites (upward black arrows) or 150,000 infected erythrocytes (upward red arrows). In keeping with the 48 h duration of *P. berghei* ANKA liver stages in C57BL/6 mice, the challenges for each pair were staggered by 48 h. For all four pairs, the sporozoite-challenged mice (green bar with red arrowhead, to indicate liver then possible erythrocytic phases) remained parasite-free at 42 days after challenge (hatched). All blood stage-challenged mice (red arrows) and placebo controls (not depicted) developed parasitemia (solid)

**Causal vs suppressive prophylaxis.** To determine whether the observed prophylaxis was causal (targeting the liver stage), suppressive (acting on erythrocytic stages) or both, paired challenges with sporozoite or erythrocytic parasites, separated by 48 h, were conducted at intervals after dosing in cohorts of three mice (Fig. 4). The erythrocytic challenge was devised to provide a bloodstream parasitemia comparable to that seen 48 h after a 5000 sporozoite inoculation. $^{ATQ}SDN_7$ protected against all sporozoite challenges but failed against all challenges by erythrocytic forms. Although a suppressive contribution may be operative after sporozoite challenge, $^{ATQ}SDN_7$ clearly provided substantial causal activity.

**Atovaquone concentrations in mouse plasma.** Atovaquone was readily detected in plasma by 6 h after intramuscular injection of

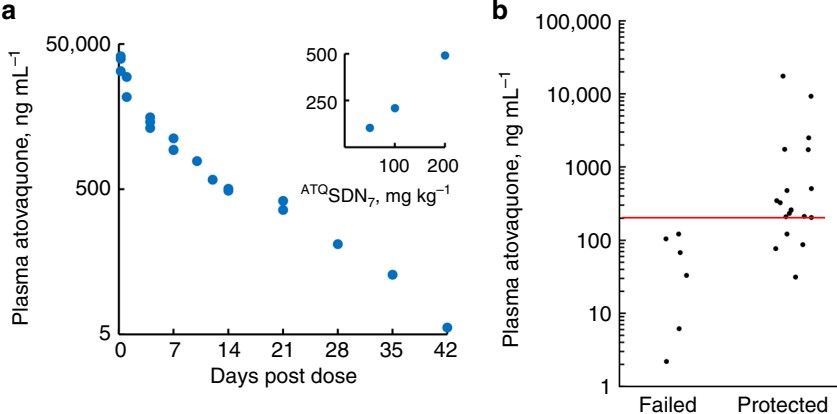

**Fig. 5** Pharmacokinetics and pharmacokinetic–pharmacodynamic relationship. **a** Plasma was collected at indicated intervals for assay of atovaquone concentrations in mice dosed intramuscularly with 200 mg kg$^{-1}$ $^{ATQ}SDN_7$. Log-transformed concentrations yield a plasma half-life of 105 h (using 4–42 days values, inclusive; $R^2$, 0.985); data obtained in six independent experiments. Insert, At 14 days after intramuscular injection, plasma concentrations (y-axis, ng mL$^{-1}$) were linear with dose (x-axis, mg kg$^{-1}$). **b** Plasma atovaquone concentrations > 200 ng mL$^{-1}$ at the time of challenge correlate well with efficacy. Each dot represents a cohort of 3–5 mice, 7–42 days after a single intramuscular dose of 36, 50, 100 or 200 mg kg$^{-1}$ $^{ATQ}SDN_7$. Data from six independent experiments

$^{ATQ}SDN_7$ (Fig. 5a) or $^{ATQ}SDN_8$ (Supplementary Fig. 6a) and levels were dose-proportional (insert, Fig. 5a). Moreover, plasma atovaquone concentrations >200 ng mL$^{-1}$ at the time of challenge were associated with efficacy of prophylaxis (Fig. 5b and Supplementary Fig. 6b). Importantly, the 105 h plasma half-life seen after day 4 was substantially longer than the 9 h seen after orally dosed atovaquone in mice[25], indicative of ongoing slow release from the depot in muscle. This is consistent with observations in humans for other LAI formulations such as rilpivirine and cabotegravir, which also exhibit a longer plasma half-life than that of their oral counterparts. The mechanisms underpinning depot release are currently poorly understood but were recently reviewed[26].

**Atovaquone resistance testing.** To evaluate whether atovaquone resistance contributes to failure after sporozoite challenge, we evaluated the atovaquone sensitivity of parasites that appeared after inadequate prophylaxis (Fig. 6). Mice were challenged 2 weeks after 36 mg kg$^{-1}$ $^{ATQ}SDN_7$, a dose and interval combination expected to fail (Fig. 3b). Erythrocytic parasites appeared, and were subinoculated into naive mice that were then treated with oral atovaquone at the lowest curative dose (Methods). Positive controls infected with atovaquone-resistant *P. berghei* became parasitemic by 2 days after oral therapy. Negative controls infected with parasites not exposed to prophylaxis were cured. Mice infected with blood from prophylaxis failures were also cured, indicating these parasites remain atovaquone-sensitive.

## Discussion

Despite promising advances toward a malaria vaccine, chemoprophylaxis remains the mainstay for malaria prevention. A safe and effective long-acting intramuscular drug-dosing preparation would provide a new tool, which joins the high efficacy of chemotherapy with the durability and ease of adherence more characteristic of a vaccine. These are, to our knowledge, the first studies of a long-acting slow-release nanoformulation of drug for causal malaria prophylaxis. They provide a convincing rationale for further exploring this strategy, and pharmacokinetic criteria for proof-of-concept clinical trial.

In the lethal *P. berghei* ANKA-C57BL/6 mouse model of malaria, $^{ATQ}SDN_7$ or $^{ATQ}SDN_8$ provided complete protection against a large sporozoite challenge for up to 4 weeks after a single

intramuscular dose (Fig. 3b). The duration of prophylaxis was dose-dependent, and comparison of plasma atovaquone levels at the time of challenge demarcates 200 ng mL$^{-1}$ as a reasonable threshold for efficacy (Fig. 5b). This benchmark is in close agreement with data from *P. falciparum* sporozoite challenge studies in non-immune healthy volunteers, in which ≥200 ng mL$^{-1}$ atovaquone at the time of challenge was at least 96% protective in 24 subjects[27, 28]. (The single failure was thought due to an assay error that led investigators to 'question whether this volunteer truly had malaria'[28].) Since the half-life after oral dosing of atovaquone in humans is allometrically eight times slower than that in mice (70 vs 9 h)[25, 27], the duration of prophylaxis afforded by a 200 mg kg$^{-1}$ slow-release depot in humans will almost certainly extend beyond the 28 days obtained in mice, at a concentration-time exposure well within the chronic >20 µg mL$^{-1}$ required in patients for *Pneumocystis* pneumonia or toxoplasmosis.

Unusual among antimalarials, atovaquone has activity against primary liver stages as well as erythrocytic parasites (causal and suppressive, respectively). When a pretreated animal is challenged with sporozoites, atovaquone thus has two opportunities against the parasite: during the initial liver stage and again during the subsequent asexual erythrocytic cycle. We were intrigued to find that in the *P. berghei* ANKA-C57BL/6 model, efficacy against sporozoite challenge is attributable largely, if not entirely, to action against hepatic forms (Fig. 4). Atovaquone levels efficacious against sporozoite challenge did not protect mice challenged with erythrocytic parasites. Though comparable relevant data in humans are limited, plasma levels substantially higher than 200 ng mL$^{-1}$ are required for activity against established erythrocytic infection[29, 30]. Furthermore, in sporozoite challenge studies, human subjects prophylaxed with atovaquone have no evidence of ever having developed an erythrocytic infection, by exquisitely sensitive PCR methods or blood culture[27, 28, 31]. Many factors could account for this apparent differential effect of atovaquone, including, for example, an intrinsically greater susceptibility of hepatic forms, drug accumulation in liver, or both.

Given the propensity for erythrocytic parasites to develop cytochrome b mutations that impart drug resistance[6, 30], we evaluated the atovaquone sensitivity of parasites that survived a 36 mg kg$^{-1}$ low prophylactic dose, when challenge was 2 weeks after injection (Fig. 6). These parasites (like their untreated controls) remained drug sensitive when subinoculated into naive

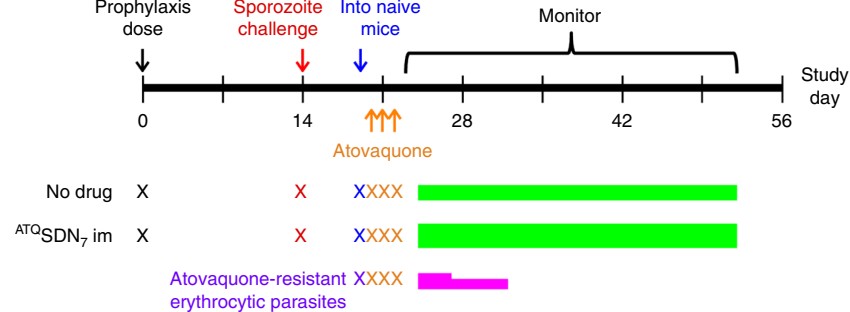

**Fig. 6** Experimental scheme and results of resistance testing. Mice treated with 36 mg kg$^{-1}$ ATQSDN$_7$ (n = 6) or no drug (n = 2) were challenged with *P. berghei* ANKA sporozoites. Erythrocytic parasites appeared in both cohorts, and were harvested and subinoculated into naive mice. At the same time a third cohort of naive mice was infected with atovaquone-resistant parasites, to serve as positive controls (n = 2). All animals were then treated orally once daily for 3 sequential days with 10 mg kg$^{-1}$ atovaquone, and monitored for parasitemia. Mice infected with parasites from the ATQSDN$_7$ prophylaxis failures, or from the no prophylaxis controls, remained parasite-free for 30 days after oral treatment (green bars). Mice infected with atovaquone-resistant *P. berghei* became parasitemic by 2 days after oral treatment and died (pink bars)

mice. This finding, in conjunction with a <120 ng mL$^{-1}$ sub-therapeutic plasma atovaquone concentration at the time of challenge, indicates therapeutic failure rather than drug resistance. At all dose levels we studied, parasitemia after intramuscular atovaquone nanoformulation was associated with plasma concentrations below 200 ng mL$^{-1}$ by the time of sporozoite challenge, consistent with a simple therapeutic breakpoint (Fig. 5b).

Although a substantial body of preclinical and clinical evidence is already available for orally dosed atovaquone, progression of this work toward clinical trial will require careful preclinical toxicity testing, especially with respect to the injection site. Our studies suggest that atovaquone alone may be suitable for malaria prophylaxis in non-immune travellers, but the obtained plasma levels and propensity for engendering resistance would proscribe its use in patients with established erythrocytic infection. Limited clinical information indicates that atovaquone may also have causal activity against vivax malaria[32], but like orally dosed atovaquone/proguanil, mefloquine or doxycycline, it would not be expected to preclude the establishment of latent hypnozoites.

Atovaquone long-acting injectable nanoformulations combine a safe, extensively studied, clinically used drug with excipients utilised in other FDA-approved medicines. Our findings suggest that a single intramuscular dose of nanoformulated atovaquone may provide causal prophylaxis against falciparum malaria for an extended period of time. This is a potential option for non-immune people travelling to malarious areas, whose trips typically last 4 weeks or less[33], and if carefully deployed it may also provide an intervention in malaria control efforts.

## Methods

**Preparation of ETFD monoliths.** Materials were purchased and used as received without further purification: α-tocopherol poly(ethylene glycol) succinate (TPGS), poly(ethylene oxide)$_{20}$ sorbitan monolaurate (Tween 20), poly(ethylene oxide)$_{20}$ sorbitan monooleate (Tween 80), sodium deoxycholate (NDC), polyvinyl alcohol (PVA), hydroxymethyl propyl cellulose (HPMC), poly(ethylene oxide)$_{101}$-block-poly(propylene oxide)$_{56}$-block-poly(ethylene oxide)$_{101}$ (F127), poly(ethylene oxide)$_{80}$-block-poly(propylene oxide)$_{27}$-block-poly(ethylene oxide)$_{80}$ (F68), and sodium 1,4-bis(2-ethylhexoxy)-1,4-dioxobutane-2-sulphonate (AOT) (Sigma-Aldrich, Dorset, UK); polyethylene glycol$^{15}$-hydroxystearate (Solutol) and polyvinyl alcohol-graft-poly(ethylene glycol) copolymer (Kollicoat) (BASF, Royal Tunbridge Wells); poly(ethylene glycol) (PEG 1k) and poly(vinylpyrrolidone) (PVP K30) (Fluka Chemicals, Dorset, UK); chloroform (Fisher Scientific, Loughbrough, UK); and atovaquone (WIS Pharmtech Co., Ltd, Shanghai, China). Statistically labelled $^3$H-atovaquone was obtained from RC Tritec Ltd (Switzerland).

**ETFD to form SDNs.** Candidate formulations were prepared using an 80 mg mL$^{-1}$ stock solution of atovaquone in chloroform, and aqueous stock solutions of 22.5 mg mL$^{-1}$ polymer and 22.5 mg mL$^{-1}$ surfactant. Stock solutions were added in the proportion of 100 μL atovaquone; 63.7 μL polymer, 31.5 μL surfactant, with 304.8 μL water, to provide a solid mass ratio of 80% atovaquone:13% polymer:7%

surfactant in an 1:4 oil:water mix. Mixtures were emulsified with a Covaris S2x for 30 s with a duty cycle of 20, intensity of 10 and 500 cycles/burst in frequency sweeping mode, after which samples were immediately cryogenically frozen. A matrix of 42 samples was prepared and lyophilised (Virtis benchtop K) for 42 h to leave a dry porous product. The samples were sealed in individual vials until analysis. $^3$H-atovaquone was added to the chloroform stock solution to generate radioactive nanoparticles of atovaquone.

For half-scale preparations, stock solutions were added at the following proportions: 50 μL atovaquone; 31.85 μL polymer, 15.75 μL surfactant, with 152.4 μL water. These were processed the same as the full scale preparations to give the same freeze-dried monoliths. For rapid equilibrium dialysis release rate studies, samples were prepared incorporating $^3$H-atovaquone at 0.2 μCi mg$^{-1}$ activity.

**Physical characterisation of ATQSDN library.** Immediately prior to analysis, samples were dispersed by addition of 16 mL of water. The Z-average diameter (nm) dispersed SDNs was measured, in triplicate, by dynamic light scattering (Malvern Zetasizer Nano ZS), using automatic measurement optimisation and Malvern Zetasizer software version 7.11 for data analysis. Eleven initial hits passed the selection criteria and demonstrated reproducibility between repeated synthesis and between full- and half-scale preparations (Supplementary Fig. 1).

**In vitro evaluation of atovaquone release rates.** The 11 candidates from physical characterisation were progressed to in vitro release kinetics using a rapid equilibrium dialysis model of an intramuscular depot. SDN samples were dispersed and diluted to 1 mg mL$^{-1}$ atovaquone in SIF, consisting of 35 mg mL$^{-1}$ bovine serum albumin (Sigma-Aldrich), 2 mg mL$^{-1}$ bovine γ-globulin (Sigma-Aldrich) dissolved in distilled water. Unformulated atovaquone was dissolved in DMSO prior to dilution with SIF, such that DMSO comprised <1% of final volume. To assess release, 0.5 mL samples were added to the donor compartment of an 8 kDa MWCO rapid equilibrium dialysis insert (ThermoFisher Scientific) and 1 mL SIF was added to the acceptor compartment. Plates containing the inserts were placed on an orbital shaker (Heidolph Rotomax 120; 100 rpm, 6 h, 37 °C) and at 0.5, 1, 2, 3, 4, 5 and 6 h, acceptor fluid (0.5 mL) was removed and replaced with fresh pre-warmed SIF. Three separate dialyses were conducted for each formulation. Aliquots (0.1 mL) of the timed samples were added to 4 mL scintillation cocktail fluid (Ultima Gold, Meridian Biotechnologies, UK) and DPM were determined (Perkin Elmer 3100TS). Data are expressed as the amount of atovaquone released and diffused across the membrane, as percent of the starting donor amount, or as a first-order release rate constant (k) calculated from the final sample.

**Dosing materials.** Dosing materials were prepared within 2 h of use and mixed thoroughly immediately prior to each injection. Nanoformulations were constituted in sterile deionized water. For oral administration, Mepron in foil sachets (GSK; Johns Hopkins Hospital pharmacy) was diluted with 10% poloxamer 188 (P5556, Sigma-Aldrich). Animals were fasted overnight prior to and 4 h after dosing.

**Ethical statement.** All animal studies were conducted under Johns Hopkins University Institutional Animal Care and Use Committee-approved protocols.

**Sporozoites and blood stage parasites for challenge.** For sporozoite challenge, 4–6 day-old female *Anopheles stephensi* mosquitoes (Liston strain) reared in the Johns Hopkins Malaria Research Institute Insectary were fasted for 6 h then fed on 6–8 week-old female Swiss Webster mice (Envigo) infected with *P. berghei* ANKA 2.34 strain (ATCC). Fed mosquitoes were maintained for 18 days (19 °C, 80%

relative humidity), when salivary glands were harvested into RPMI medium and disrupted by several passages through a 0.5 in, 28-gauge needle to release sporozoites. Parasites were counted by hemocytometer and diluted to 25,000 mL$^{-1}$ RPMI. For blood stage challenge, donor mice (6-week-old C57BL/6 male mice, Jackson Laboratory) were infected by tail vein infusion of 5000 *P. berghei* ANKA sporozoites, and at ≥0.5% parasitemia were anaesthetised (intraperitoneal 100 mg kg$^{-1}$ ketamine, 5 mg kg$^{-1}$ acepromazine). Blood was harvested by cardiac puncture and diluted with RPMI to obtain $1.5 \times 10^5$ infected erythrocytes per 200 uL.

**In vivo efficacy and pharmacokinetics**. All studies were conducted with *P. berghei* (ANKA 2.34 strain) in 6-week-old C57BL/6 male mice (Jackson Laboratory). Animals were dosed just once and challenged just once. Mice (~20 g) were injected intramuscularly (biceps femoris; 27-gauge needle, 100 μL Hamilton syringe, part number 7656–01) with the test nanoformulation or vehicle (20 μL per limb, 40 μL total volume). Oral atovaquone was administered by gavage. At predetermined intervals after dosing, mice were challenged by tail vein infusion of 5000 sporozoites, or $1.5 \times 10^5$ infected erythrocytes, in 200 μL RPMI. Starting 4 days after challenge then weekly thereafter for 42 days, tail snip blood samples were examined for parasites via Giemsa-stained thin smears. A sample was deemed negative if no parasites were seen in 2000 erythrocytes (limit of detection, 0.05% parasitemia). Animals were monitored daily for overall morbidity and mortality. Each experiment had a vehicle control to validate challenges. The efficacy outcome was binary: protected or failed. Treatment was deemed protective if all animals in at least two independent experiments remained parasite-free at 42 days post challenge. Any partial protection (e.g., delayed parasitemia or just one unprotected mouse in a cohort) was nevertheless deemed a failure. Blood for pharmacokinetics was harvested (microtainer tubes, BD Biosciences), centrifuged (1300×g, 10 min, 4 °C), and plasma was collected and stored at −80 °C until use.

**Generation of atovaquone-resistant erythrocytic P. berghei**. Six-week-old C57BL/6 male mice (Jackson Laboratory) were inoculated intraperitoneally with 150,000 infected erythrocytes, and at ~3% parasitemia cohorts of 3–6 mice were treated daily for 3 days with oral doses of 1, 10, 100, or 1000 mg kg$^{-1}$ atovaquone. Mice in the 10, 100 and 1000 mg kg$^{-1}$ groups remained parasite-free at 30 days post infection, and 10 mg kg$^{-1}$ was designated the lowest curative dose. Mice treated with three daily doses of 1 mg kg$^{-1}$ recrudesced 9 days after the final dose and were then given a single oral dose of 10 mg kg$^{-1}$ atovaquone. Parasitemia fell transiently then began increasing on day 3, when blood was harvested, pooled, and 10$^6$ parasitized cells per animal were subinoculated into naive mice. Cohorts of two subinoculated mice were treated daily for 3 days with oral doses of 1, 10, or 100 mg kg$^{-1}$ atovaquone. All animals recrudesced. On day 8 blood from mice that failed 100 mg kg$^{-1}$ was harvested, pooled, diluted 1:1 in Alsever's solution with 10% glycerol, stored at −80 °C, and used as the resistance reference line.

**Atovaquone sensitivity of parasites from animals that failed prophylaxis**. Six-week-old C57BL/6 male mice (Jackson Laboratory) dosed on study day 0 with 36 mg kg$^{-1}$ intramuscular $^{ATQ}SDN_7$ (n = 6) or no drug (n = 2) were challenged on day 14 with 5000 sporozoites. On day 19 blood was harvested from parasitemic animals in the treated (n = 3) or placebo groups (n = 2), pooled accordingly, diluted with phosphate buffered saline, and subinoculated intraperitoneally into naive mice at 10$^6$ parasites per mouse (three treated, two control). For a positive control, naive mice (n = 2) were inoculated intraperitoneally with 10$^6$ atovaquone-resistant erythrocytic parasites (above). All subinoculated mice were treated once daily on study days 20, 21 and 22, with an oral dose of 10 mg kg$^{-1}$ atovaquone, the lowest curative regimen (above), then monitored 30 days for parasitemia.

**Atovaquone concentration in plasma**. Atovaquone concentration in plasma was determined by UPLC-MS/MS, using a modification of a previously described method that was validated to FDA standards[34]. In brief, acetonitrile and deuterated atovaquone internal standard (Toronto Research Chemicals) were added to 25 μL thawed plasma, followed by vortex mixing and centrifugation to obtain the supernatant. A 10 μL aliquot was separated by linear gradient on 50 × 2 mm, 2.5 μm Polar-RP 100 A column (Synergi), and atovaquone was monitored by triple-quadrupole API 4000 mass analyser with electrospray ionisation (SCIEX, Framingham, MA, USA). The assay was validated for 5–5000 ng mL$^{-1}$ and interpolations were by $1/x^2$-weighted least-squares-fitted quadratic regression. All 77 reported values were determined by this method. A subset of 42 samples, analysed in another laboratory by the previous method[34], agreed within 15%.

**Statistical analyses**. Descriptive values in the text are M ± SD. Plasma half-life was estimated by linear regression of log concentration on time in a sparse sampling non-compartmental analysis using Phoenix WinNonlin 8.0 (Certara, Princeton, NJ, USA). Fisher's exact test was used to compare the pooled proportions of untreated mice and mice treated with successful regimens (for each dose-to-challenge interval), using Stata 14.0 (StataCorp, College Station, TX, USA).

**Data availability**. The authors confirm that all relevant data are available upon request.

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

## Acknowledgements

We thank Charles Flexner for linking this collaboration; David Sullivan for thoughtful discussions; Emily Caton and Mary Barry for laboratory support; Teri Parsons and Mark Marzinke for assistance with atovaquone assays; and Professor Gary Posner for generously gifting LC-MS instrumentation. We are grateful to the Insect and Parasite Core Facilities of the Johns Hopkins Malaria Research Institute for mosquito rearing and provision of sporozoites. This work was supported by a Johns Hopkins Malaria Research Institute Pilot grant and by the Bloomberg Philanthropies (to T.A.S.), by R01AI095453 (to T.A.S.) for atovaquone assay development, and by fellowship support from the National Institutes of Health (T32GM066691) and the Sherrilyn and Ken Fisher Center for Environmental Infectious Diseases (to M.M.I.). Infrastructure and methods employed in this work were developed with support from the Engineering and Physical Science Research Council (EP/G066272/1, EP/K002201/1, EP/L02635X/1) and the National Institutes of Health (R24AI118397, R01AI114405-01) (to A.O. and S.P.R.).

## Author contributions

A.S., A.O., S.P.R. designed and manufactured the nanoformulation library and screened physico-chemical outcomes to select candidates. L.T., A.O., S.P.R. determined in vitro drug-release rates and selected nanoformulations for progression to in vivo studies. R.P.B., A.K.T., G.M., T.A.S. designed all in vivo studies. R.P.B., A.K.T., T.A.S. dosed and obtained blood samples; A.K.T. assessed parasitemia; A.K.T., G.M. conducted challenges; E.N. developed and conducted assays of atovaquone in plasma; M.M.I. did pharmacokinetic and statistical analyses. R.P.B., A.K.T., G.M., E.N., M.M.I., T.A.S. analysed and interpreted in vivo data. All authors contributed to writing of the paper.

## Additional information

**Competing interests:** The authors declare no competing financial interests. S.P.R., A.O., A.C.S., L.T.,T.A.S., R.P.B., G.M. and A.K.T. are inventors on a patent filing describing the use of atovaquone SDNs.

