## [Peer Review File · Nature Communications]

Reviewers' comments:

Reviewer #1 (Remarks to the Author):

This is a clearly written and well formatted paper utilizing a library building strategy to formulate atovaquone nanoparticles for use as long-acting injectables (LAI) to act as a malaria prophylaxis for travelers.

Context is important in research related to malaria. Combating malaria has a long history where prevention, diagnosis and treatment have all been intensively researched; and progress has been made. However there is no denying that malaria is a devastating disease that poses incalculable harm to people and the economies they live within. While vaccines may become available for endemic populations (e.g. RTS,S), travelers do need (and already have) access the prophylaxis. Many options already exist. Malarone (atovaquone/proguanil) is already a widely used oral malaria prophylaxis (one tablet/day during travel; start 1-2 d before travel and continue for 7 d after travel). Malarone dose regimen can be tailored to travel time and the drug can be stopped if the patient suffers hypersensitivity-this is not possible with long acting version of atovaquone. Monoprophylaxis/therapy may result in relapse of *P. vivax* malaria, so it is not clear if the recent findings (Ref 5, line 62) that resistant *P. falciparum* cannot survive in mosquitoes is also relevant for *P. vivax* or other forms of malaria. I do not understand the rationale that while atovaquone resistant *P. falciparum* cannot survive in mosquitoes, how this would help protect a traveler. What happens if a traveler is infected and *P. falciparum* becomes resistant in the traveler? Just because some mosquitoes may not be able to pass resistant *P. falciparum*, there will be other forms of malaria present and a large/diverse mosquito population. I think the manuscript under plays the complexity of the situation and a general reader should have a more information. Would atovaquone mono-prophylaxis work to provide causal prophylaxis against other forms of malaria (e.g. *P. vivax*)? I think the manuscript would profit by introducing these issues more frankly, i.e. (1) Malarone dosing regimen-widely used, simple, flexible, (2) risk of mono therapy for other forms of malaria and (3) as a prophylaxis how would mono-atovaquone be better positioned to protect a traveler compared to existing prophylaxis (in addition to malarone). I think highlighting this context is important for moderating the conclusion (lines 158-160).

Line 32. Please indicate what is the novel strategy. The Authors have published the strategy (Ref 17) to use existing excipients to make libraries of solid nanoparticles of other drug active substances in a similar vein as described in this manuscript. Emulsion templated freeze drying is well known. Applying a known strategy to a different drug substance is not a novel strategy.

Lines 124-125. Please clearly indicate the route of administration as being intramuscular. For the general reader some comment about the difference between intramuscular and subcutaneous routes of administration would be helpful. An explanation for the use of intramuscular over subcutaneous would be helpful. It is not clear from the manuscript why intramuscular is the desired route of administration. If

Line 128 Does oral bioavailability of atovaquone in the mouse compared to humans?

Line 132. For the sake of the general reader, please define causal and suppressive prophylaxis.

Line 144. I am not sure that '7-d half-life' is the correct term for the observation of atovaquone from a depot. C_{max} was not determined. I suspect that as depots, LAIs dissolution and diffusion of the drug active substance into blood does not change the drug half life once drug is in the blood. Increased bioavailability is due to the depot effect of the formulation, not an increase in drug half life.

Line 150. How does a dose of >200 ng/mL prophylaxis of atovaquone compare to what is known for blood levels with the prophylaxis provided by Malarone for example? What is known about the correlation of the observed efficacy of atovaquone blood levels in the mouse compared to the human for atovaquone alone and in combination?

Line 148+ and Line 153+. The authors state the underpinning release of LAIs is not well understood (line 148) and then say humans and mice can be allometrically correlated (line 153). It is not clear how these two statements can be reconciled.

Are there significant differences in the data shown in extended data Figures 2, 3 and 4?

Please describe in the methods section how atovaquone loading was determined.

Please comment on the use of *P. berghei* in the in vivo study. It should be clear in the manuscript that *P. berghei* was used and to the extent how the use of this parasite might compare to *P. falciparum* in humans. I would suggest this point is made clear in the conclusion.

Reviewer #2 (Remarks to the Author):

The authors present a long acting injectable formulation of a well-known antimalarial, atovaquone, as monotherapy. The approach is novel and may be useful to avoid compliance issues, specially for travelers and patients in endemic areas with access to the appropriate healthcare settings, if efficacy and safety of the new formulation are demonstrated.

There are some experiments that will strengthen your conclusions:

-Antimalarial chemotherapy is only recommended in combination to avoid selection of resistant mutants. In your work, atovaquone is administered alone without its usual partner, proguanil. Considerable evidence suggests that malaria parasites become resistant to atovaquone quickly if atovaquone is used as a sole agent (frequency of resistance ≤ 1 in 10⁵ parasites).

In the efficacy experiments presented in this work, parasitemia appears at all the doses administered of the LAI formulation (week 1.5, 4 or 5 depending on the dose). From the

results presented, is not clear whether these parasites are resistant mutants, or correspond to a therapeutic failure. To clarify this, it would be very useful to inject those “potential” resistant parasites to naïve mice and demonstrate whether they are susceptible to atovaquone (therapeutic failure) or resistant mutants.

-Back to back comparison with atovaquone administered by oral route in terms of efficacy and toxicity should be shown to demonstrate that the new formulation is better or at least equivalent to the original drug. Atovaquone administered by oral route as a single agent has a relatively favorable side effect profile due to its poor bioavailability. Using this formulation atovaquone is being administered by intramuscular route. It would be needed to provide data to determine the safety profile of this long-acting formulation that will maintain higher levels of the drug during prolonged period (1 month in humans).

Although the *Plasmodium falciparum* mouse model is expensive, it is accessible through collaborations with MMV or the BMGF. It would add value to your article to test this new formulation in the real human parasite.

Specific comments:

Line 124. Please indicate the route of administration and the doses used for atovaquone, ATQSDN7 and ATQSDN8. It is stated that there is some protection when given one day before the challenge. Could you clarify the level of protection? Need to clarify the level of protection when the formulations were administered 2 or 4 days before the challenge (according to Fig2a)

Line 126. The level of protection provided by several doses of ATQSDN7 and ATQSDN8 and atovaquone were assessed for up to 6 weeks. Need to indicate the method used to measure parasitemia and limit of detection of this methodology (it is not described in Mat and Met). Need to clarify the statistics: How many replicates were done? How many animals were used per group?

Parasitemia is shown in Figure 3b as circles or x. Need to indicate what these symbols represent, e.g. the mean of several experiments. Need to include SD or error bars.

Line 127 and Figure 3b. According to figure 3b doses of 36, 50, 100 and 200mg/kg were protective during 1, 3 and 4 weeks respectively, but after this time, parasitemia reappear in all the groups tested. Need to clarify whether these parasites correspond to recrudescence or resistant mutants.

One of the main issues of atovaquone is the high rate of spontaneous resistance. It would add value to your article to determine the frequency of resistance of atovaquone as monotherapy when is used in this novel formulation.

Line 136. Need to indicate the dose at which ATQSDN7 was efficacious. In figure 3c please indicate the day at which parasitemia is detected in the mice, it would be very useful for the reader.

Line 138. Atovaquone is efficacious against *P.falciparum* and *P.bergei* erythrocytic stages, both in vitro (in the case of *P.falciparum*) and in animal models of infection. Please explain why atovaquone LAI formulations are not efficacious against erythrocytic forms.

Line 143. Need to indicate the range of doses of the different pharmacokinetics conducted to state that atovaquone concentrations were dose-proportional after administration of ATQSDN7.

Line 150. Please remove the term “successful prophylaxis” since after administration of the different doses of LAI, parasitemia has reappeared. A term indicating the duration of

protection would be more appropriate.

Leyend. Figure 3a. Please indicate that figure 3a is the scheme of the experimental procedure. It would be useful for the reader.

Leyend. Figure 3b. Need to indicate number of mice per group, method used to determine parasitemia. Please indicate if the dots are the mean of different experiments, include SD or error bars

Leyend. Figure 3c. Please indicate that the figure is the scheme of the experimental procedure

Leyend. Figure 3d. Need to indicate the replicates conducted, number of animals and show error bars.

Leyend. Figure 3d (insert). Need to indicate the doses tested

Materials and Methods. Efficacy chapter. Need to indicate the method used to determine the level of parasitemia in the mice, and the limit of detection of that technique.

Reviewer #3 (Remarks to the Author):

The manuscript describes formulation of atovaquone nano drug delivery systems. Manuscript is well written. The topic, a long acting formulation for malaria prophylaxis is an important topic in medical research. Data is technically sound and conclusions are based on the research data.

Minor points related to the research plan, which should be commented, are:

- 1) Chloroform was used as a solvent for preparation of nano drug delivery systems. How it was confirmed that the level of solvent residues were acceptable in the final formulation?
- 2) What is the reasoning for 6 hour duration of in vitro drug release tests? The time span is very short and does not have relevance for long acting once per month intramuscular drug delivery system?
- 3) In the dialysis drug release testing, how it was checked that the drug is not interacting with the dialysis membrane and that the drug penetration through the membrane is not the rate limiting step for drug release (this is a very common problem with the dialysis membranes, and it should be commented in the text)?

Point-by-point response to reviewers

Verbatim reviewer comments are italicized and numbered. Corresponding author replies are not italicized. Cited literature is listed at the end.

Reviewer #1

1-1. Many options already exist. Malarone (atovaquone/proguanil) is already a widely used oral malaria prophylaxis (one tablet/day during travel; start 1-2 d before travel and continue for 7 d after travel). Malarone dose regimen can be tailored to travel time and the drug can be stopped if the patient suffers hypersensitivity- this is not possible with long acting version of atovaquone.

Inability to stop an intramuscular dose is a real concern, for any drug. Nevertheless, there are dozens of agents (e.g., anti-psychotics, antibiotics, contraceptives, opioids, corticosteroids, vitamins), including formulations designed specifically to be long-acting, which are dosed intramuscularly. This is despite the availability of cognate oral formulations, and attests to the fact that benefits of intramuscular dosing can outweigh risks. Intramuscular dosing is often chosen for its greater convenience and/or compliance, both key considerations for malaria prophylaxis. The literature is replete with reports of the commonplace incomplete compliance with oral malaria chemoprophylaxis (typically ~50%), even with current drug of choice Malarone [1-3]. Given that the risk of malaria is up to 3.4% per month in unprotected travelers [4], and that malaria morbidity and mortality is especially severe in nonimmunes, the advantage of intramuscular dosing in this context is considerable. After 25 years in clinical use atovaquone is regarded as safe, only rarely associated with severe toxicity despite its (confounding) use in immunocompromised and ill patients (atovaquone alone), or, for malaria, its obligate coadministration with proguanil, itself having toxicities.

*1-2. Monoprophylaxis/therapy may result in relapse of *P. vivax* malaria, so it is not clear if the recent findings (Ref 5, line 62) that resistant *P. falciparum* cannot survive in mosquitoes is also relevant for *P. vivax* or other forms of malaria. I do not understand the rationale that while atovaquone resistant *P. falciparum* cannot survive in mosquitoes, how this would help protect a traveler. What happens if a traveler is infected and *P. falciparum* becomes resistant in the traveler? Just because some mosquitoes may not be able to pass resistant *P. falciparum*, there will be other forms of malaria present and a large/diverse mosquito population.*

On a worldwide basis vivax malaria accounts for 4% of malaria cases and 0.7% of deaths [5]. Prevalence and disproportionate morbidity and mortality make falciparum malaria by far the most important species for prophylaxis, and vaccine programs are, accordingly, overwhelmingly directed against *P. falciparum*. Nevertheless, *P. vivax* is the predominant species in many malarious areas outside of Africa, and any prophylactic measure would ideally cover it (as well as *ovale*, *malariae* and *knowlesi*).

We can find no publication bearing on atovaquone monoprophylaxis (or even monotherapy) for *P. vivax* infection. The available information for atovaquone/proguanil indicates there is reliable activity against erythrocytic vivax parasites (84-100%) and poor activity against hypnozoites [6-10]. Interestingly, there is also the suggestion of meaningful action against the initial liver stages of vivax (causal activity), with atovaquone/proguanil being noninferior to primaquine in uninfected nonimmune humans [10,11]. We find no report on atovaquone-resistant vivax parasites in mosquitoes, and these studies would be extraordinarily difficult to do. By any dosing route, prophylaxis by atovaquone, with or without proguanil, would not be expected to preclude recurrent parasitemia from hypnozoites.

Prophylaxis by intramuscular atovaquone would only be reliable if administered to people who are free of erythrocytic parasites at the time of dosing. In that scenario, therapeutic levels of drug will already be present at the time of sporozoite inoculation, and, because resistant parasites cannot survive in mosquitoes, any introduced parasites must be atovaquone-sensitive. When atovaquone monotherapy is given to patients with erythrocytic infection (parasite burden up to 10^{12}) drug-resistant cytochrome b mutants are readily generated and selected [6]. As now explicitly stated in the text, intramuscular atovaquone monotherapy would therefore be inappropriate for treatment of established malaria infection. We do not know whether liver stages can similarly tolerate cytochrome b mutations – the mosquito stages cannot. In any case, our new subinoculation studies give no indication of resistance after sporozoite challenge (new Fig. 6). For atovaquone alone in nonimmune humans, activity against liver stage *P. falciparum* is seen at 20-fold lower drug exposures than are required for erythrocytic parasites [12,13]. Causal potency for atovaquone/proguanil is comparable to that of atovaquone alone [14]. Given these different sensitivities, and the substantially smaller number of parasites

that infect the liver (just dozens), it seems unlikely that in a pre-treated person liver stage parasites would survive to establish a viable erythrocytic infection, capable of becoming drug resistant. Supporting this reasoning, a careful analysis of atovaquone resistance in erythrocytic parasites indicates that the mutation rate is 10^{-5} or less [25]. Since amplification in liver is 10^4 -fold, this threshold is unfavorable for resistance in liver stages.

Available data obviously do not fully address these important questions, and they cannot be answered by lab studies, only by thoughtful clinical trials.

1-3. Would atovaquone mono-prophylaxis work to provide causal prophylaxis against other forms of malaria (e.g. P vivax)?

Please see above reply to 1-2. In vitro data indicate atovaquone has activity against primary liver stage *P. cynomolgi* (a surrogate for *P. vivax*) but not the hypnozoite forms [26].

1-4. moderating the conclusion (lines 158-160).

The manuscript has been extensively edited to reflect this concern.

1-5. Line 32. Please indicate what is the novel strategy. The Authors have published the strategy (Ref 17) to use existing excipients to make libraries of solid nanoparticles of other drug active substances in a similar vein as described in this manuscript. Emulsion templated freeze drying is well known. Applying a known strategy to a different drug substance is not a novel strategy.

There are a number of ways in which the data and strategy within the current manuscript are novel and novelty is explicitly exemplified by our successful patent application. The novelty is further summarized as follows:

- a) Emulsion-templated freeze drying has not previously been applied to generating a long-acting injectable formulation for any molecule for any indication.
- b) No previous strategy has unequivocally (or otherwise) demonstrated pharmacokinetic exposure and pharmacodynamic efficacy for a period of 28-days in animals for malaria prophylaxis or therapy.
- c) As the reviewer correctly states, the specific technology has not been successfully applied to the formation of solid drug nanoparticles for atovaquone, or any other antimalarial agent.

1-6. Lines 124-125. Please clearly indicate the route of administration as being intramuscular. For the general reader some comment about the difference between intramuscular and subcutaneous routes of administration would be helpful. An explanation for the use of intramuscular over subcutaneous would be helpful. It is not clear from the manuscript why intramuscular is the desired route of administration. If

The manuscript has been revised to clarify routes of administration as being oral (controls only) or intramuscular (nanoformulations). Prospectively we could find no clear rationale in the literature for intramuscular vs subcutaneous dosing. It is generally agreed that in humans the volume limit for subcutaneous is 1 mL (vs 3-5 mL for muscle), and that subcutaneous dosing may deliver higher drug levels to the lymphatic system, which would be important for anti-retrovirals but not for malaria. We chose the intramuscular route because it accommodates larger possible volumes.

1-7. Line 128 Does oral bioavailability of atovaquone in the mouse compared to humans?

The oral bioavailability of atovaquone is poor in mice, rats, rabbits, dogs and humans, ranging from 5-51%, and is particularly affected by formulation, fasted/fed status, and fat content of coadministered food [15,16]. To minimize variability, we fasted mice before and after dosing, and dosed orally with Mepron, the commercial product with formulation optimized for oral dosing. See 1-10 below for mouse and human blood levels.

1-8. Line 132. For the sake of the general reader, please define causal and suppressive prophylaxis.

This was done in the legend to Fig. 1, but is now incorporated into the text as well.

1-9. Line 144. I am not sure that '7-d half-life' is the correct term for the observation of atovaquone from a depot. Cmax was not determined. I suspect that as depots, LAIs dissolution and diffusion of the drug active substance into blood does not change the drug half life once drug is in the blood. Increased bioavailability is due to the depot effect of the formulation, not an increase in drug half life.

We now use the purely descriptive term "plasma half-life" for the falling levels over time in plasma, and make clear in the text that it obviously reflects ongoing release from the depot.

1-10. Line 150. How does a dose of >200 ng/mL prophylaxis of atovaquone compare to what is known for blood levels with the prophylaxis provided by Malarone for example? What is known about the correlation of the observed efficacy of atovaquone blood levels in the mouse compared to the human for atovaquone alone and in combination?

See also above reply to 1-2. Plasma levels in mice dosed intramuscularly with 200 mg kg⁻¹ nanoformulated atovaquone are depicted in new Fig. 5a, with a threshold for prophylaxis at 200 ng mL⁻¹ (new Fig. 5b). When given with or without proguanil, atovaquone in humans is similarly active against *P. falciparum* liver stages at >200 ng mL⁻¹ [12,14]. For atovaquone given with proguanil at daily recommended prophylaxis dose, steady state in humans is reached by day 3 [17] with chronic trough levels of 2 µg mL⁻¹ [18]. It should be noted that the recommended prophylaxis regimen, which results in greater than necessary atovaquone levels, was devised to accommodate the considerably shorter half-life of proguanil. It also reflects the sensible marketing of just one tablet composition, to serve both prophylaxis (which includes a causal component) and treatment (directed against the less sensitive and resistance-prone red cell forms, and with no causal intent).

There is very little information on the antimalarial PK-PD of atovaquone in vivo, in the literature or in FDA's online Summary Basis for Approval [19]. We know of no reports, other than our manuscript, that relate plasma atovaquone levels to antimalarial activity in mice. Our observed 200 ng mL⁻¹ threshold for causal activity against *P. berghei* correlates well with that for *P. falciparum* in humans [12,14]. Similarly, our finding that concentrations >200 ng mL⁻¹ fail to suppressively prophylax against *P. berghei* parallels the inability of atovaquone alone to cure erythrocytic falciparum infection in nonimmune patients with peak plasma levels up to 3.4 µg mL⁻¹ [13], and it's 60% fail rate in endemic patients [6] treated with doses that in other studies yielded steady state levels of 14 µg mL⁻¹ [20]. Interpretation of the reported activity of atovaquone alone against erythrocytic parasites in patients must be tempered by the (incompletely or not described) confounder of drug resistance generated in this lifecycle stage [6,13].

1-11. Line 148+ and Line 153+. The authors state the underpinning release of LAIs is not well understood (line 148) and then say humans and mice can be allometrically correlated (line 153). It is not clear how these two statements can be reconciled.

The statement regarding allometric scaling within the manuscript clearly refers to the half-life for orally administered atovaquone. However, the authors acknowledge that the use of the word "disposition" within this sentence was potentially misleading. Therefore, the sentence has been reworded to clarify the intention "Since the *half-life after oral dosing* of atovaquone in humans is allometrically eight times slower than that in mice....".

1-12. Are there significant differences in the data shown in extended data Figures 2, 3 and 4?

As already described within the paper, three distinct solid drug nanoparticle formulations were studied and these were designated ^{ATQ}SDN₄, ^{ATQ}SDN₆, and ^{ATQ}SDN₈. Extended data Figures 2, 3 and 4 present data at different drug loading for ^{ATQ}SDN₄, ^{ATQ}SDN₆, and ^{ATQ}SDN₈, respectively. This is already defined within the figure legends of these figures.

1-13. Please describe in the methods section how atovaquone loading was determined.

The inclusion of solid, non-volatile materials to the emulsion prior to freeze drying leads to the same total mass of the solids after removal of all volatile components after free-drying (as would be expected). Furthermore, the radiometric studies described in the methods section directly measures the atovaquone loading and this is implicit within the text of the original submission. We do not see any need to further describe the loading determination as this would constitute duplication within the manuscript.

1-14. Please comment on the use of *P. berghei* in the in vivo study. It should be clear in the manuscript that *P. berghei* was used and to the extent how the use of this parasite might compare to *P. falciparum* in humans. I would suggest this point is made clear in the conclusion.

The text has been expanded to provide background on the *P. berghei* ANKA-C57BL/6 system. No murine model is 100% faithful to *P. falciparum* in humans, particularly with respect to cycle times (duration in liver, length of red cell cycle). Despite its already-reported 48 h duration and because of its importance to this

project, as described previously we carefully measured the length of exoerythrocytic stage of this model, so as to have confidence in its value in our hands, and to use it for design of the causal/suppressive experiment (new Fig. 4). *P. berghei* ANKA-C57BL/6 is commonly used in drug development, and has a number of key characteristics that make it a valuable pre-clinical model of falciparum: all lifecycle forms are represented, there are no latent hypnozoite forms, drug sensitivities are comparable to those of *P. falciparum* (except for species-specific targets, such as some of the plasmepsins, a concern not relevant for phylogenetically conserved atovaquone target cytochrome b), and of considerable importance, *P. berghei* ANKA parasitemia in C57BL/6 is progressive and reliably lethal unless drug-cured [21,22]. Challenge by 5000 *P. berghei* ANKA sporozoites in C57BL/6 poses a stringent test for chemoprophylaxis (50 are enough to establish reliable parasitemia [23]), and drug failure results in unremitting parasitemia that kills within days. Since there are no self-cures, partial treatments soon become evident. Our 6 wk follow-up monitoring period exceeds by 3 weeks the most delayed appearance of parasitemia (21 days), so partial protection is highly unlikely to have gone undetected.

Reviewer #2

2-1. In the efficacy experiments presented in this work, parasitemia appears at all the doses administered of the LAI formulation (week 1.5, 4 or 5 depending on the dose). From the results presented, is not clear whether these parasites are resistant mutants, or correspond to a therapeutic failure. To clarify this, it would be very useful to inject those "potential" resistant parasites to naïve mice and demonstrate whether they are susceptible to atovaquone (therapeutic failure) or resistant mutants.

We regret the brevity and lack of clarity in writing that led to this conclusion, which is incorrect. Parasitemia never developed in those animals deemed "protected" (green dots in former Fig. 3b), and we monitored for 6 weeks after challenge to be confident in this endpoint. We believed all failures resulted from sub-therapeutic drug concentrations, a conclusion supported by the PK data. Nevertheless, because resistance is of particular concern for atovaquone therapy, we conducted the suggested subinoculation experiment and the revision now includes an analysis of atovaquone susceptibility in parasites that emerge when challenge is 2 weeks after an intramuscular dose of 36 mg kg⁻¹ (new Fig. 6). Atovaquone sensitivity of the subinoculated parasites confirms that failure is indeed attributable to inadequate drug levels, not resistance.

2-2. Back to back comparison with atovaquone administered by oral route in terms of efficacy and toxicity should be shown to demonstrate that the new formulation is better or at least equivalent to the original drug. Atovaquone administered by oral route as a single agent has a relatively favorable side effect profile due to its poor bioavailability. Using this formulation atovaquone is being administered by intramuscular route. It would be needed to provide data to determine the safety profile of this long-acting formulation that will maintain higher levels of the drug during prolonged period (1 month in humans).

The tolerance and safety of orally dosed atovaquone (alone or with proguanil) has been extensively documented in several animal species and in humans (clinical trials and 25 years' postmarketing experience). Steady state plasma concentrations up to 50 µg mL⁻¹ for several weeks were well-tolerated in humans [21]. FDA-approved use of atovaquone for chronic prophylaxis against *Pneumocystis pneumonia* is dosed to sustain the required plasma levels of ≥20 µg mL⁻¹ [24]. Since causal antimalarial activity in humans is seen at one-tenth this level (0.2 µg mL⁻¹, see also reply to 2.9 below), the concentration-time exposure for malaria prophylaxis is expected to be well within that already in clinical use for other indications.

2-3. Although the Plasmodium falciparum mouse model is expensive, it is accessible through collaborations with MMV or the BMGF. It would add value to your article to test this new formulation in the real human parasite.

Additional information with the human pathogen could certainly strengthen the case. However, the questions we are asking would require mice dually engrafted with both human liver tissue and erythrocytes. Such animals have been prepared, but they are not yet well characterized with respect to response to standard antimalarials, they are variable between individuals in the ratio of mouse/human liver tissue, available only in small numbers, and severely immunodeficient. They do not survive unless some mouse liver tissue is present, which could confound both PK and PD aspects of our work. Finally, we are concerned that drug release rate from an intramuscular injection may vary meaningfully from that in a normal mouse, where inflammation/healing affect bioavailability.

Specific comments:

2-4. Line 124. Please indicate the route of administration and the doses used for atovaquone, ATQSDN7 and ATQSDN8. It is stated that there is some protection when given one day before the challenge. Could you clarify the level of protection? Need to clarify the level of protection when the formulations were administered 2 or 4 days before the challenge (according to Fig2a)

The manuscript has been extensively revised and has several new figures, so as to address this and multiple related comments. In summary, the nanoformulations were given by intramuscular route only, and oral controls were with Mepron (see also reply to 1-7). New Fig. 3b depicts the durations (dose-to-challenge intervals) and efficacies of protection. Candidate nanoformulations were initially screened by a one day postdose challenge, which three of the five contenders failed. The one day challenge experiment was not repeated for ^{ATQ}SDN₇ or ^{ATQ}SDN₈, and so is not reported in Fig. 3b (which includes only results obtained from at least two independent experiments). Regarding the final sentence of this comment, the X-axis in old Fig. 3a was in weeks not days. We did no challenges at 2 or 4 days after dosing. New Figs. 3a and b are clearer on these points.

2-5. Line 126. The level of protection provided by several doses of ATQSDN7 and ATQSDN8 and atovaquone were assessed for up to 6 weeks. Need to indicate the method used to measure parasitemia and limit of detection of this methodology (it is not described in Mat and Met).

Need to clarify the statistics: How many replicates were done? How many animals were used per group?

As previously described in lines 8 and 9 of the efficacy section of Methods, and retained in the revision, the endpoint of these studies was parasitemia in tail snip blood samples, examined by Giemsa-stained thin smear. As now stated, a sample was deemed negative if no parasites were seen in 2000 erythrocytes (limit of detection 0.05% parasitemia). This method was chosen because *P. berghei* ANKA is inexorable and inevitably fatal in C57BL/6 mice (see also above reply to 1-14), and ample time after challenge was allowed (6 weeks) for even one drug-surviving parasite to become detectable (in fact, fatally so). Replicates and statistics for Fig. 3b are now explicit.

2-6. Parasitemia is shown in Figure 3b as circles or x. Need to indicate what these symbols represent, e.g. the mean of several experiments. Need to include SD or error bars.

Fig. 3b, its legend, and the general text have been thoroughly revised to resolve these issues.

2-7. Line 127 and Figure 3b. According to figure 3b doses of 36, 50, 100 and 200mg/kg were protective during 1, 3 and 4 weeks respectively, but after this time, parasitemia reappear in all the groups tested. Need to clarify whether these parasites correspond to recrudescence or resistant mutants. One of the main issues of atovaquone is the high rate of spontaneous resistance. It would add value to your article to determine the frequency of resistance of atovaquone as monotherapy when is used in this novel formulation.

Please see the above reply to 2-1. There was no parasitemia, for up to six weeks after challenge, in any of the mice in cohorts deemed successfully protected (green dots in old Fig. 3b). Nevertheless, as suggested we have now included the suggested subinoculation study assessing atovaquone sensitivity of erythrocytic parasites that appear after failed prophylaxis regimens. These parasites remain atovaquone sensitive (new Fig. 6) and they occur in animals whose plasma levels at the time of challenge fall below the 200 ng mL⁻¹ threshold for causal efficacy (new Fig. 5b). These multiple lines of evidence support the notion that the prophylaxis failures we see are attributable to subtherapeutic doses/concentrations and not to resistance.

2-8. Line 136. Need to indicate the dose at which ATQSDN7 was efficacious. In figure 3c please indicate the day at which parasitemia is detected in the mice, it would be very useful for the reader.

As for several of the above issues, the previous manuscript did not adequately describe experimental design and results. There was no parasitemia in animals deemed "protected". Extensive revisions have been made to format, text and figures so as to remedy this miscommunication.

2-9. Line 138. Atovaquone is efficacious against *P.falciparum* and *P.berghei* erythrocytic stages, both in vitro (in the case of *P.falciparum*) and in animal models of infection. Please explain why atovaquone LAI formulations are not efficacious against erythrocytic forms.

See also above replies to 1-2 and 1-10; only limited PK-PD data are available for atovaquone in vivo and we find no reports other than ours that bear on plasma levels in mouse malaria. Atovaquone has potent subnanomolar activity against *P. falciparum* parasites in vitro. A substantial contribution to the disparity between in vitro/in vivo effective concentrations is attributable to protein binding, which exceeds 99%. For atovaquone alone in human *P. falciparum*, causal activity is seen at total plasma levels $>200 \text{ ng mL}^{-1}$ (550 nM) [12], whereas activity against erythrocytic parasites is incomplete at $3\text{-}14 \mu\text{g mL}^{-1}$ (8-38 μM) [6,13]. Our mouse studies do not identify a threshold for activity against erythrocytic *P. berghei* in mice, but the finding that liver stages are more sensitive than erythrocytic ones parallels the data for *falciparum* in humans. As now mentioned in the manuscript, many factors could contribute to better activity against liver stages, including for example greater intrinsic susceptibility of the liver stages to atovaquone, or higher tissue levels of drug in liver than in erythrocytes.

2-10. Line 143. Need to indicate the range of doses of the different pharmacokinetics conducted to state that atovaquone concentrations were dose-proportional after administration of ATQSDN7.

This is now explicit in the insert and legend to new Fig. 5a

2-11. Line 150. Please remove the term "successful prophylaxis" since after administration of the different doses of LAI, parasitemia has reappeared. A term indicating the duration of protection would be more appropriate.

See also the above replies to 2-1, 2-7, and 2-8. We saw no parasitemia, for up to 6 weeks after challenge, in treatments deemed "successful prophylaxis" and have therefore retained this terminology. The text and figures have been extensively modified to clarify the data.

2-12. Legend. Figure 3a. Please indicate that figure 3a is the scheme of the experimental procedure. It would be useful for the reader.

New Fig. 3a has been simplified and the legend revised to indicate it is an experimental scheme.

2-13. Legend. Figure 3b. Need to indicate number of mice per group, method used to determine parasitemia. Please indicate if the dots are the mean of different experiments, include SD or error bars

This figure and its legend have been extensively revised to provide this information.

2-14. Legend. Figure 3c. Please indicate that the figure is the scheme of the experimental procedure

We have edited the legend of new Fig. 4 to clarify that it provides both experimental scheme and results.

2-15. Legend. Figure 3d. Need to indicate the replicates conducted, number of animals and show error bars.

The legend to new Fig. 5a addresses these matters. Error bars have not been provided because values are (variably) from 1, 2 or 3 replicates, and because the data are remarkably well-behaved (R^2 0.985 for kinetics over a 38 d interval), despite the fact that a) each blood sample was taken from a different mouse (hence all variability is interindividual), b) samples are from six independent experiments over time, and c) dosing is by technically demanding injection into the small muscle of a mouse. Furthermore, as now also noted in Methods, 42 of the 77 PK samples were analyzed by two different LC-MS methods in two different laboratories, and concordance was within 15%. The new analytical method provides a 50-fold increase in sensitivity.

2-16. Legend. Figure 3d (insert). Need to indicate the doses tested

Dose is indicated on the x-axis of the insert.

2-17. Materials and Methods. Efficacy chapter. Need to indicate the method used to determine the level of parasitemia in the mice, and the limit of detection of that technique.

Please see above replies to 1-14 and 2-5.

Reviewer #3

3-1. Chloroform was used as a solvent for preparation of nano drug delivery systems. How it was confirmed that the level of solvent residues were acceptable in the final formulation?

Chloroform is a class 2 solvent as directed by the ICH guidelines for residual solvents with a concentration limit of 60 ppm. In numerous studies with freeze-drying of chloroform and dichloromethane emulsions using the emulsion-templating approach to form nanoparticles, we have conducted GC-headspace and weight determinations (in some cases under cGMP conditions) and we have regularly achieved concentrations below the ICH guidelines (including 600ppm for dichloromethane). In this preclinical study we have not conducted such studies as the use of radioactivity precludes access to a number of facilities that are not licensed to use radioactive compounds. The later non-radioactive versions used in this proof-of-concept study were produced under identical conditions to the radioactive samples leading to selection of hits. The absolute study of residual solvents would form part of later studies to translate these materials to clinical candidates.

3-2. What is the reasoning for 6 hour duration of in vitro drug release tests? The time span is very short and does not have relevance for long acting once per month intramuscular drug delivery system?

The authors acknowledge the concern of the reviewer regarding the timing for the in vitro selection. However, the presented assay was previously validated against clinical pharmacokinetic data for clinically available long-acting formulations for schizophrenia (paliperidone palmitate, risperidone), and contraception (medroxyprogesterone acetate), and found to display a good in vitro - in vivo correlation. The key premise of the assay is that it measures drug release kinetics while sink conditions are fulfilled in vitro, which are likely to be maintained over the in vivo dosing interval. Notably, for this reason it is common in other in vitro systems such as Caco-2 cells to measure apparent permeability at 1 hour for predicting oral bioavailability, when actual drug absorption in vivo occurs over a much longer period of time. Importantly, the caco-2 cell assay is widely applied in all major pharmaceutical companies during drug development despite this. The success of our in vitro selection strategy is also further exemplified within the current manuscript through the accelerated identification of an atovaquone formulation providing 28-day prophylactic exposure, without having to characterize the entire formulation space in vivo.

3-3. In the dialysis drug release testing, how it was checked that the drug is not interacting with the dialysis membrane and that the drug penetration through the membrane is not the rate limiting step for drug release (this is very common problem with the dialysis membranes, and it should be commented in the text)?

Again, the authors acknowledge the concern of the reviewer regarding the interaction of the drug with the dialysis membrane, and agree that this is a common problem with this type of assay. We refer the editor to our response to point 3-2 above regarding assay validation, which is also pertinent to this issue. While the interaction of atovaquone (or any other agent) cannot be completely discounted in any in vitro assay involving the use of plastics, it has been minimized within our optimized assay via the constituents of the buffers used. Moreover, the data presented within the manuscript clearly demonstrate remarkable differences in drug release kinetics using this assay, which can only be attributed to differences in formulation because atovaquone is common to all of them. Therefore, while some interaction of the atovaquone with the membrane is inevitable there is no basis for this not being consistent between formulations, and it is the relative difference in measured release kinetics between formulations that has been used as a basis for lead selection.

Literature Cited in Point-by-Point

1. Mace KE, Arguin PM; Centers for Disease Control and Prevention (CDC) (2017) Malaria Surveillance - United States, 2014. *MMWR Surveill Summ* 66:1-25
2. Créach AM, Velut G, deLaval F, Briolant S, Aigle L, Marimoutou C, Deparis X, Meynard J, Pradines B, Simon F, Michel R, Mayet A (2016) factors associated with malaria chemoprophylaxis compliance among French service members deployed in Central African Republic. *Malar J* 15:174-83
3. Saunders DL, Garges E, Manning JE, Bennett K, Schaffer S, Kosmowski AJ, Magill AJ (2015) Safety, tolerability, and compliance with long-term antimalarial chemoprophylaxis in American soldiers in Afghanistan. *Am J Trop Med Hyg* 93:584-90
4. Freedman DO, Chen LH, Kozarsky PE (2016) Medical considerations before international travel. *N Engl J Med*. 375:247-60
5. WHO (2017) World Health Organization Fact Sheet: World Malaria Report 2016. (2016)
6. Looareesuwan S, Viravan C, Webster HK, Kyle DE, Hutchinson DB, Canfield CJ. Clinical studies of atovaquone, alone or in combination with other antimalarial drugs, for treatment of acute uncomplicated malaria in Thailand. *Am J Trop Med Hyg*. 1996 Jan;54(1):62-6.
7. Looareesuwan S, Wilairatana P, Glanarongran R, Indravijit KA, Supeeranontha L, Chinnapha S, Scott TR, Chulay JD. Atovaquone and proguanil hydrochloride followed by primaquine for treatment of Plasmodium vivax malaria in Thailand. *Trans R Soc Trop Med Hyg*. 1999 Nov-Dec;93(6):637-40.
8. Ling J, Baird JK, Fryauff DJ, Sismadi P, Bangs MJ, Lacy M, Barcus MJ, Gramzinski R, Maguire JD, Kumusumangsih M, Miller GB, Jones TR, Chulay JD, Hoffman SL; Naval Medical Research Unit 2 Clinical Trial Team. Randomized, placebo-controlled trial of atovaquone/proguanil for the prevention of Plasmodium falciparum or Plasmodium vivax malaria among migrants to Papua, Indonesia. *Clin Infect Dis*. 2002 Oct 1;35(7):825-33.
9. Soto J, Toledo J, Luzz M, Gutierrez P, Berman J, Duparc S. Randomized, double-blind, placebo-controlled study of Malarone for malaria prophylaxis in non-immune Colombian soldiers. *Am J Trop Med Hyg*. 2006 Sep;75(3):430-3.
10. Kolifarhood G, Raeisi A, Ranjbar M, Haghdoost AA, Schapira A, Hashemi S, Masoumi-Asl H, Mozafar Saadati H, Azimi S, Khosravi N, Kondrashin A. Prophylactic efficacy of primaquine for preventing Plasmodium falciparum and Plasmodium vivax parasitaemia in travelers: A meta-analysis and systematic review. *Travel Med Infect Dis*. 2017 May - Jun;17:5-18.
11. Mavrogordato A, Lever AM. A cluster of Plasmodium vivax malaria in an expedition group to Ethiopia: prophylactic efficacy of atovaquone/proguanil on liver stages of P. vivax. *J Infect*. 2012 Sep;65(3):269-74.
12. Shapiro, T. A., Ranasinha, C. D., Kumar, N. & Barditch-Crovo, P. Prophylactic activity of atovaquone against Plasmodium falciparum in humans. *Am J Trop Med Hyg* 60, 831-836 (1999).
13. Chiodini PL, Conlon CP, Hutchinson DB, Farquhar JA, Hall AP, Peto TE, Birley H, Warrell DA (1995) Evaluation of atovaquone in the treatment of patients with uncomplicated Plasmodium falciparum malaria. *J Antimicrob Chemother* 36:1073-8.
14. Deye GA, Miller RS, Miller L, Salas CJ, Tosh D, Macareo L, Smith BL, Francisco S, Clemens EG, Murphy J, Sousa JC, Dumler JS, Magill AJ (2012) Prolonged protection provided by a single dose of atovaquone-proguanil for the chemoprophylaxis of Plasmodium falciparum malaria in a human challenge model. *Clin Infect Dis* 54:232-9.
15. Pudney M, Gutteridge W, Zeman A, Dickins M, Woolley JL (1999) Atovaquone and proguanil hydrochloride: a review of nonclinical studies. *J Travel Med* 6 Suppl 1:S8-12.
16. CDER/FDA Clinical Pharmacology/Biopharmaceutics Review Mepron NDA#020500/S005 16Dec1998 (https://www.accessdata.fda.gov/drugsatfda_docs/nda/99/020500s005_chemr_micror_clinphmr.pdf)
17. Berman JD, Nielsen R, Chulay JD, Dowler M, Kain KC, Kester KE, Williams J, Whelen AC, Shmuklarsky MJ (2001) Causal prophylactic efficacy of atovaquone-proguanil (Malarone) in a human challenge model. *Trans R Soc Trop Med Hyg* 95:429-32
18. Sukwa TY, Mulenga M, Chisdaka N, Roskell NS, Scott TR (1999) A randomized, double-blind, placebo-controlled field trial to determine the efficacy and safety of Malarone (atovaquone/proguanil) for the prophylaxis of malaria in Zambia. *Am J Trop Med Hyg* 60:521-5
19. US FDA (1999) NDA 21-078 Clinical Pharmacology and Biopharmaceutics Reviews (http://www.accessdata.fda.gov/drugsatfda_docs/nda/2000/21-078_Malarone_BioPharmr.pdf)

20. Hughes WT, Kennedy W, Shenep JL, Flynn PM, Hetherington SV, Fullen G, Lancaster DJ, Stein DS, Palte S, Rosenbaum D, San HT, Liao M, Blum R, Rogers MD (1991) Safety and pharmacokinetics of 566C80, a hydroxynaphthoquinone with anti-Pneumocystis carinii activity: A Phase I study in human immunodeficiency virus (HIV)-infected men. *J Infect Dis* 163:843-8
21. Matuschewski K (2013) Murine infection models for vaccine development: The malaria example. *Hum Vaccin Immunother* 9:450-6
22. Bagot S, Idrissa Boubou M, Campino S, Behrschmidt C, Gorgette O, Guénet JL, Penha-Gonçalves C, Mazier D, Pied S, Cazenave PA Bagot S(1), Idrissa Boubou M, Campino S, Behrschmidt C, Gorgette O, Guénet JL, Penha-Gonçalves C, Mazier D, Pied S, Cazenave PA (2002) Susceptibility to experimental cerebral malaria induced by Plasmodium berghei ANKA in inbred mouse strains recently derived from wild stock. *Infect Immun* 70:2049-56
23. Scheller LF, Wirtz RA, Azad AF (1994) Susceptibility of different strains of mice to hepatic infection with Plasmodium-berghei. *Infect Immun* 62:4844-7
24. GlaxoSmithKline Inc. Product Monograph: Mepron. 26July2016 (<https://ca.gsk.com/media/591391/mepron.pdf>)
25. Gassis S, Rathod P (1996) Frequency of drug resistance in Plasmodium falciparum: Nonsynergistic combination of 5-fluoroorotate and atovaquone suppresses in vitro resistance. *Antimicrob Agents Chemother* 40:914-9
26. Dembele L, Gego A, Zeeman AM, Franetich JF, Silvie O, Rametti A, Le Grand R, Dereuddre-Bosquet N, Sauerwein R, van Gemert GJ, Vaillant JC, Thomas AW, Snounou G, Kocken CH, Mazier D (2011) Towards an in vitro model of Plasmodium hypnozoites suitable for drug discovery. *PLoS One* 6:e18162

REVIEWERS' COMMENTS:

Reviewer #1 (Remarks to the Author):

Overall response to reviewer's comments appear detailed and most of the revisions to the manuscript appear sound. I recommend publication.

Minor points

(1-4) I still feel the conclusion (line 264+) is overstating what is possible for the atovaquone formulation as prophylaxis against falciparum. I leave this to the Editors to decide if more qualification of the results is necessary in line 266.

(1.5) I acknowledge most of the Authors comments in 1-5 and leave to the Editors if the application described in this manuscript is the best demonstration for an extended PK/PD example as stated by the Authors for novelty. I would disagree that a patent application exemplifies novelty. Patent grant is still required, and much else besides (e.g. optimizing, scaling, registration etc etc). A complex parenteral prophylaxis would need to compete with current prophylaxis regimens (some very efficient) and other advances in the battle against malaria.

(1-13) I still cannot find an actual absolute loading of atovaquone relative to excipients in material injected into mice. I will not persist on this point that this information be added by delaying publication. I understand the response regarding radiometric studies and see that if the data were easily available it would have been added. Confirmation of loading seems a straight forward experiment. I would have thought that loading confirmation was done in cold experiments to optimize the process. I understand many processes exist where recovery 'would be expected' to be quantitative, but confirmation is generally recommended.

Reviewer #3 (Remarks to the Author):

The concerns of the reviewers have been taken into account and responded in a satisfactory way.